# A Yellow Fever Virus 17D Infection and Disease Mouse Model Used to Evaluate a Chimeric Binjari-Yellow Fever Virus Vaccine

**DOI:** 10.3390/vaccines8030368

**Published:** 2020-07-09

**Authors:** Kexin Yan, Laura J. Vet, Bing Tang, Jody Hobson-Peters, Daniel J. Rawle, Thuy T. Le, Thibaut Larcher, Roy A. Hall, Andreas Suhrbier

**Affiliations:** 1Inflammation Biology Group, QIMR Berghofer Medical Research Institute, Brisbane 4029, Australia; Kexin.Yan@qimrberghofer.edu.au (K.Y.); Bing.Tang@qimrberghofer.edu.au (B.T.); Daniel.Rawle@qimrberghofer.edu.au (D.J.R.); Thuy.Le@qimrberghofer.edu.au (T.T.L.); 2School of Chemistry and Molecular Biosciences, University of Queensland, St Lucia 4072, Australia; l.vet@uq.edu.au (L.J.V.); j.peters2@uq.edu.au (J.H.-P.); 3Australian Infectious Disease Research Centre, Brisbane 4029, Australia; 4Institut National de Recherche Agronomique, Unité Mixte de Recherche 703, Oniris, Nantes CEDEX 03, France; thibaut.larcher@inrae.fr

**Keywords:** yellow fever virus, mouse model, vaccine, Binjari virus

## Abstract

Despite the availability of an effective, live attenuated yellow fever virus (YFV) vaccine (YFV 17D), this flavivirus still causes up to ≈60,000 deaths annually. A number of new approaches are seeking to address vaccine supply issues and improve safety for the immunocompromised vaccine recipients. Herein we describe an adult female IFNAR-/- mouse model of YFV 17D infection and disease that recapitulates many features of infection and disease in humans. We used this model to evaluate a new YFV vaccine that is based on a recently described chimeric Binjari virus (BinJV) vaccine technology. BinJV is an insect-specific flavivirus and the chimeric YFV vaccine (BinJ/YFV-prME) was generated by replacing the prME genes of BinJV with the prME genes of YFV 17D. Such BinJV chimeras retain their ability to replicate to high titers in C6/36 mosquito cells (allowing vaccine production), but are unable to replicate in vertebrate cells. Vaccination with adjuvanted BinJ/YFV-prME induced neutralizing antibodies and protected mice against infection, weight loss and liver pathology after YFV 17D challenge.

## 1. Introduction

Yellow fever virus (YFV) is endemic in 47 countries in Africa and Central and South America. During 2013 the virus caused 84,000–170,000 severe cases and 29,000–60,000 deaths globally [1]. Large outbreaks have occurred in recent years in Angola, Democratic Republic of Congo and Brazil [2]. A revised strategy to Eliminate Yellow Fever Epidemics (EYE) was launched by the World Health Organization (WHO) in 2017, with affordable vaccines and a sustained vaccine market representing key objectives [3,4]. Yellow fever (YF) is prevented by an effective live attenuated vaccine (17D), with a single dose able to confer life-long protection in most people. The global demand for YF vaccine supply has been estimated to be ≈102 million doses in 2016, and is expected to rise to ≈140 million doses in 2021 [5]. WHO has licensed four YFV vaccine strains that are used in the production of YF vaccines manufactured by six companies [6], who together produce an annual volume of around 80–90 million doses. Although there are three YFV lineages there appears to be only one serotype [6]. The recent shortage of YF vaccine may soon be addressed as Sanofi Pasteur transitions to a new production facility [7]. In the interim, fractional dosing has been shown to be effective [8]. The YF vaccine is given as a single subcutaneous or intramuscular injection, with some countries requiring an International Certificate of Vaccination against YF for incoming visitors.

The current YF vaccines comprise the live attenuated virus (YFV 17D) that was generated by serial passage of the pathogenic YFV strain, Asibi (isolated in Ghana in 1927), in mouse embryo and chicken tissue culture [9,10,11]. YF vaccines are manufactured in specific pathogen free (SPF) embryonated chicken eggs, which are also used in the manufacture of influenza vaccines. Transition to cell culture-based influenza vaccine manufacture (e.g., Celvapan) is considered important for mitigating against influenza vaccine shortages in pandemic situations by reducing the reliance on SPF egg supply [12]. YF vaccines (like most influenza vaccines) are also contraindicated in persons with a history of anaphylactic reactions to egg proteins. Future YF vaccines might thus similarly seek not to be reliant on SPF eggs. Because YF vaccines are live-attenuated, they are contraindicated in immunocompromised individuals (e.g., those on immunosuppressant medication and HIV-infected persons), infants between zero and six months old, individuals over 59 years of age, pregnant women and nursing mothers. Severe side effects outside such individuals are generally rare and include nervous system reactions (≈1 in 125,000) and life threatening severe illness with organ failure (≈1 in 250,000) [13,14,15].

The aforementioned issues associated with the current YFV 17D vaccines has spawned a number of approaches to develop new YF vaccine strategies [6,16]. These include production of the live attenuated vaccine in Vero cells (ClinicalTrials.gov Identifier: NCT04142086), a beta-propiolactone inactivated YF vaccine [17,18] (ClinicalTrials.gov Identifier: NCT00995865), a Modified Vaccinia Ankara vectored recombinant YF vaccine [19] (ClinicalTrials.gov Identifier: NCT02743455) and (at a preclinical stage) a virus-like particle (VLP) YF vaccine manufactured in transfected HEK293 cells [20].

The structural proteins of flaviviruses comprise two proteins, envelope (E) and membrane (M), with the latter synthesized as a precursor (prM), collectively referred to as prME [21]. Neutralizing antibodies directed at E are believed to represent the primary mediators of protection for most flavivirus vaccines, including YFV vaccines [22,23]. We recently described a new chimeric flavivirus vaccine platform based on the insect specific flavivirus Binjari virus [24,25]. Chimeric viruses are generated by replacing the prME genes of Binjari virus with the prME genes of the target pathogenic flavivirus, with the resulting chimeras encoding the non-structural proteins and capsid genes from Binjari virus and the prME genes of the pathogenic flavivirus. Such Binjari chimeras replicate to high titers in C6/36 cells (a mosquito larva-derived cell line), generating structurally authentic flavivirus virons that contain the prME proteins of the pathogenic flavivirus. These chimeras are unable to replicate in vaccine recipients, as the non-structural proteins of Binjari virus are defective for viral RNA replication in vertebrate cells [24]. The platform was used to generate a BinJ/YFV-prME vaccine using the prME sequence of YFV 17D-204 [24].

Wild-type YFV is classified as a biosafety level 3 (BSL3) organism, whereas culture and growth of the YFV 17D vaccine strain is classified as a BSL2 activity. Mouse models have been developed using YFV 17D including (i) infection of AG129 mice resulting in neurotropic disease [26] and (ii) infection of young (3–4 week old) IFNAR-/- mice showing viscerotropic and neurotropic disease [27], although Meier et al., reported no pathology in 3–4 week old A129 mice infected with YFV 17D-204 [28]. Herein we describe an adult IFNAR-/- mouse model of YFV 17D-204 (hereafter referred to as YFV 17D) infection and disease, which is suitable for pre-clinical evaluation of YFV vaccines. The model shows viraemia, weight loss and liver pathology, and was used to evaluate protection mediated by an adjuvanted BinJ/YFV-prME vaccine.

## 2. Materials and Methods

### 2.1. Ethics Statement

All mouse work was conducted in accordance with the “Australian code for the care and use of animals for scientific purposes” as defined by the National Health and Medical Research Council of Australia. Mouse work was approved by the QIMR Berghofer Medical Research Institute animal ethics committee (P2195).

### 2.2. Mice, Infection, Virus Titration, and Liver Preparation

IFNAR-/- mice were kindly provided by P. Hertzog (Monash University, Melbourne, Australia) and were on a C57BL/6J background [29] and were bred in-house at QIMR B [30]. YFV 17D [10,11] was propagated in low passage C6/36 cells (ATCC CRL-1660), with both virus and cells negative for mycoplasma [31], with low endotoxin fetal calf serum [32] used for culture. Mice were infected with 5 log_10_CCID_50_ of YFV 17D subcutaneously at the base of the tail in 100 µL. Serum was collected into Microvette Serum-Gel 500 µL blood collection tubes (Sarstedt, Numbrecht, Germany) via tail nicks. Viraemias were determined by CCID_50_ assays, as described previously [33,34]. Briefly, serum was tittered in duplicate using 10-fold serial dilutions on C6/36 cells (5 day culture) followed by parallel well-to-well transfer of supernatants into 96 well plates containing Vero E6 cells (C1008, ECACC, Wiltshire, England; Sigma Aldridge, St. Louis, MO, USA). After 5 days of culture, cytopathic effects indicate virus positive wells.

Mice were weighed daily and on day 6 post infection mice were euthanized with CO_2_ and liver lobes were cut into large pieces and fixed in paraformaldehyde (for paraffin sections, H&E and immunohistochemistry) and some placed in RNAlater (Qiagen, Hilden, Germany) (for qRT PCR).

### 2.3. The Vaccine

The BinJ/YFV-prME vaccine was constructed, and produced in C6/36 cells, as described previously [24]. For production, C6/36 cells were infected at a MOI ≈ 0.1 followed by culture for 5 days, supernatants were harvested, with cell debris removed by 30 min spin at 2000× *g* at 4 °C, polyethylene glycol 8000 (40% PEG8000 w/v in 10 mM Tris pH 8, 1 mM EDTA and 120 mM NaCl) was then added to virus supernatant (1:4 vol:vol) with overnight rotation at 4 °C. The solution was then ultra-centrifuged (SW32Ti, Beckman, Brea, CA, USA) (average 18,000× *g*) for 2 h, with the supernatant removed and the virus precipitate suspended in 10 mM Tris pH 8, 1 mM EDTA and 120 mM NaCl. This solution was underlaid with a ≈5 mL 20% sucrose cushion (in 10 mM Tris pH 8, 120 mM NaCl) and centrifuged (average 100,000× *g*) for 2 h at 4 °C. The supernatant and sucrose were discarded and the pellet resuspended in 10 mM Tris pH 8, 120 mM NaCl (buffer), and assayed for protein content using Pierce BCA protein Assay (Thermo Scientific, Scoresby, Australia) and analyzed by SDS PAGE.

The adjuvant comprised QS-21 (50 µg/mL) (Creative Biolabs, Shirley, NY, USA), 3-O-desacyl-4’-monophosphoryl lipid A (MPLA) (50 µg/mL), cholesterol (250 µg/mL), dioleoyl phosphatidylcholine (1 mg/mL) (Sigma-Aldrich, St. Louis, MO, USA) constituted in PBS by sonication. BinJ/YFV-prME (at the indicated doses per mouse) was mixed with adjuvant (1:1 vol:vol) with 40 µL injected i.m. into each quadriceps muscle such that the total mouse dose of QS-21 (and MPLA) was 2 µg per mouse per time point for both priming and boosting rounds of injections. Two control groups comprised mice injected as above with Buffer, and mice injected as above with buffer mixed with adjuvant.

### 2.4. Real Time Quantitative RT-PCR (qRT-PCR)

Liver lobes were cut in half lengthways stored in RNAlater solution, kept overnight at 4 °C and then stored at −80 °C. RNA was extracted by thawing the liver pieces in TRIzol (Life Technologies, Carlsbad, CA, USA) while homogenized using four ceramic beads at 6000 rpm twice for 15 s (Precellys 24 Homogeniser, Bertin Instruments, Montigny-le-Bretonneux, France). cDNA was generated using iScript cDNA Synthesis kit (Bio-Rad, Irvine, CA, USA) according to the manufacturer’s instructions. qPCR was performed using the following primers: YFV 17D Forward 5ʹ GTATTCTGTGGATGCTGACC 3ʹ, YFV 17D Reverse 5ʹ TATCCCGGTTTCAGGTTGTG 3ʹ, which span the YFV NS5-3’UTR junction [35], with normalization using RPL13A cDNA levels as described [36]. qPCR was performed in a reaction consisting of 2 µL of cDNA, 5 µL of SYBR green Super mix-UDG (Invitrogen, Carlsbad, CA, United States), 2 µL water, and 1 µL of 10 mM of forward and reverse primers. The qPCR reaction was undertaken using CFX 96 touch PCR detection system (Bio-Rad) under the following cycling conditions: 1 × 95 °C 3 min, 40 × 95 °C 5 s and 60 °C 30 s. Data were analysed using Bio-Rad CFX Real Time Analysis software (Bio-Rad, Irvine, CA, USA). Data is presented as a ratio of YFV 17D RNA levels over RPL13A mRNA levels, with levels quantitated using a standard curve of serially diluted positive sample.

### 2.5. Neutralization Assay

YFV neutralizing antibody titers were determined (against YFV-17D) by incubating 2-fold serial dilutions of heat-inactivated mouse serum (in duplicate, 50 µL) with 4 log_10_CCID_50_ of virus (50 µL) for 1 h before adding to Vero E6 cells (ECACC Vero C1008, Sigma Aldridge, St. Louis, MO, USA), 10^4^ cells/well in a 96-well plate in 100 µL medium comprising RPMI1640 supplemented with 2% FBS. After 5 days incubation at 37 °C and 5% CO_2_, cells were fixed and stained with formaldehyde (7.5% w/v)/crystal violet (0.05% w/v) for 0.5–1 h, washed twice in water, plates dried overnight, 100 µL of 100% cold methanol added per well and OD_595_ read. The reciprocal 50% neutralizing titer was determined by linear interpolation of OD values using Excel (Microsoft, Redmont, WA, USA), with 0% neutralization set by wells with virus and no anti-serum, and 100% neutralization set by wells with no virus and no anti-serum.

### 2.6. Histology and Immunohistochemistry

Liver lobes were cut and fixed in 10% formalin overnight, embedded in paraffin and sections stained with hematoxylin and eosin (H & E). For immunohistochemistry, paraffin sections were rehydrated via an alcohol series, placed in Target Retrieval Solution, pH 9.0 (Dako/Agilent, Santa Clara, CA, USA) for 20 min at 100 °C using the Decloaking chamber (Biocare Medical, Pacheco, CA USA), blocked using Rodent Block M (Biocare Medical) overnight and Biocare Medical Background Sniper (with 1% BSA, 20% donkey serum, 20% goat serum) for 15 min, and then stained overnight at room temperature with anti-flavivirus NS1 monoclonal antibody 4G4 (Mozzy mAbs; https://eshop.uniquest.com.au/mozzy-mabs/) as tissue culture supernatant diluted in Da Vinci Green Diluent (Biocare Medical). After washing, endogenous peroxidase was blocked by incubation in 2% hydrogen peroxide for 10 min and ImmPRESS HRP (horse) Anti-Mouse IgG (Peroxidase) Polymer Detection Kit (Vector Laboratories, Burlingame, CA, USA) added for 1 h. After washing, signal was developed using Vector NovaRED Peroxidase (HRP) Substrate Kit (LS Bio, Seattle, WA, USA), with light haematoxylin counter-staining.

### 2.7. Statistics

Statistical analysis of experimental data was performed using IBM SPSS Statistics for Windows, Version 19.0 (IBM Corp., Armonk, NY, USA). The t-test was used when the data was deemed parametric i.e. difference in variances was <4, skewness was >2 and kurtosis was <2. Otherwise the non-parametric Kolmogorov-Smirnov test was used.

## 3. Results

### 3.1. Infection and Disease Mediated by The YFV 17D Vaccine Strain in Adult IFNAR-/- Mice

Male and female IFNAR-/- mice were infected s.c. with 5 log_10_CCID_50_ of YFV 17D. A 4–5 day viraemia ensued, peaking at about 4 log_10_CCID_50_/mL (Figure 1a). This was accompanied by a significant weight loss from day 2 to day 6 (Figure 1b, *p* < 0.003). No significant gender differences were apparent.

Mice were euthanized on day 6, with overt white necrotic lesions visible on livers of infected female mice (Figure 2a), but not uninfected female mice (Figure 2b). For 29 infected female IFNAR-/- mice, 48% manifested this phenotype. The mean age of mice with these lesions was 13.5 SD 2.7 (range 10–19 weeks) and those without was 19 SD 12 (range 10–55 weeks), with the difference not reaching significance (*p* = 0.43, Kolmogorov Smirnov test). Of the small number of female mice with age >20 weeks (*n* = 4), none manifested these liver lesions. Of 17 males tested, none showed such lesions (mean age 15.6 SD 3.3, range 12–20 weeks).

The necrotic lesions were clearly evident by H & E staining of liver sections (Figure 2c). The periphery of such lesions also stained for flaviviral antigen by immunohistochemistry (IHC), indicating viral infection of hepatocytes (Figure 2d), an observation consistent with studies in YF patients [37,38]. H & E staining also showed the presence of Councilman bodies (range of 2–5 per liver section) (Figure 2f), a histological feature originally deemed pathognomonic for YFV infection [38,39]. Some YFV 17D-infected mice also showed clear signs of steatosis (abnormal retention of lipid within hepatocytes) (Figure 2g, small white inclusions), a feature well described in YF patients [37].

The reason for the name yellow fever is the presence of bilirubin in the blood which gives the skin a yellow appearance. However, bilirubin levels were not elevated in any of the mice tested day 6 post infection. Elevated levels of alkaline phosphatase (ALP) and aspartate transaminase (ASP) [40] were seen in some mice at this time, indicating reduced liver function, although levels did not clearly correlate with the presence of the overt liver lesions nor were there significant differences between male and female mice (Appendix A). Both ALT and/or ASP can be elevated in a hamster YFV model [41] and in YF patients [42,43].

### 3.2. BinJ/YFV-prME Vaccination and Challenge in Adult Female IFNAR-/- Mice

The chimeric BinJ/YFV-prME virus (Appendix A) was generated as described [24], was grown in C6/36 cells and was purified using a sucrose cushion, with SDS PAGE of the final preparation clearly showing the structural proteins (Appendix A). Although the BinJ/YFV-prME chimera comprises a fully formed virus that is replication competent in C6/36 cells, it might be viewed as more of a VLP-like vaccine given that it is unable to replicate in vertebrate cells [24]. VLP-based vaccines are generally adjuvanted, with AS01 used in two VLP-based vaccines; a malaria vaccine (Mosquirix) and a tetravalent human papillomavirus VLP vaccine [44]. AS01 is also used in the licensed varicella zoster virus vaccine, Shingrix. The composition of this adjuvant and the dose used in mice has been published [45,46]. An in-house version of this adjuvant was produced and comprised the TLR4 agonist, 3-O-desacyl-4’-monophosphoryl lipid A and the saponin QS-21, with both adjuvant components used at 2 µg per mouse for both prime and boost [45,46] (Figure 3a).

Female IFNAR-/- mice were vaccinated twice with 5, 10 or 20 µg of BinJ/YFV-prME formulated with or without the aforementioned adjuvant (Figure 3a). Neutralizing antibodies were detected after two vaccinations with adjuvanted BinJ/YFV-prME (Figure 3b). No neutralizing responses were detected (limit of detection 1 in 10 dilution of serum) after 2 vaccinations with unadjuvanted vaccine or Buffer (Figure 3b). As expected, convalescent sera from YFV 17D-infected IFNAR-/- mice showed high levels of neutralizing antibody responses (Figure 3b, CS).

In two previous experiments we saw no or minimal protection against YFV 17D-mediated infection and disease in mice that had no detectable neutralizing antibodies, consistent with the prevailing view that neutralizing antibodies represent the primary correlate of protection [47]. The 5 µg and 20 µg plus adjuvant groups were thus selected for challenge with YFV 17D (Figure 3a). Two control groups were included, Buffer and Buffer with adjuvant. After challenge, the vaccinated mice showed no detectable viraemia, whereas the two control groups showed similar levels of viraemia (Figure 3c). Both vaccine groups also showed no weight loss, whereas both control groups showed significant (Figure 3d, days 3–6, *p* < 0.01) weight loss, although adjuvant did provide some protection against weight loss on days 5 and 6 (Figure 3d). qRT PCR for YFV 17D in livers showed significantly higher levels of viral RNA in the control groups when compared with the vaccinated groups (Figure 3e).

### 3.3. Post Challenge Liver Histology

Mouse livers taken day 6 post challenge with YFV 17D (and from naïve mice) were examined by histology, with H & E staining used to identify necrotic lesions (Figure 2c), inflammatory infiltrate foci (Figure 2d; Appendix A), Councilman bodies (Figure 2f) and steatosis (Figure 2g). The overt white necrotic lesions (Figure 2a,c) were present in 2/6 and 3/6 mice in the control groups, but absent from the BinJ/YFV-prME + adjuvant and Naïve groups (Table 1). Councilman bodies were present in 5/6 and 6/6 mice in the control groups, but absent from the BinJ/YFV-prME + adjuvant and Naïve groups (Table 1). Steatosis (a relatively non-specific manifestation) was also absent from vaccinated and naïve mice (Table 1). Also present in all the livers were foci of cellular infiltrates (Table 1; Appendix A). The mean number of these foci in control groups was 17.3 SD 8.1 and 17.3 SD 8.6, whereas they were significantly lower in both the 5 µg (8 SD 2.4) and 20 µg (4.8 SD 0.98) vaccine groups (all *p* = 0.031, using the non-parametric Kolmogorov Smirnov test as differences in variance were >4). Vaccination thus significantly reduced the number of these liver lesions.

There were 2.6-fold more foci of cellular infiltrates (4.8 SD 0.98) in the 20 µg vaccine group than in the Naive group (1.8 SD 1.2) (Table 1), which reached significance (*p* = 0.031). There were also 1.7 fold more such lesions in the 5 µg vaccine group, when compared with the 20 µg vaccine group (*p* = 0.031, Kolmogorov Smirnov tests). These results suggest that vaccination did not provide sterilizing immunity as some level of liver inflammation occurred even in the 20 µg vaccine group. Although such foci of inflammatory infiltrates are seen at low levels in naïve mice (Table 1), the increased numbers in infected mice likely arise from infiltrates forming around foci of viral infection (Figure 2d; Appendix A).

## 4. Discussion

We describe herein a mouse model of YFV 17D infection and disease, which shows a number of similarities to human infection and disease but results in minimal welfare issues for infected mice. Use of the YFV 17D vaccine strain permits experiments under BSL2 containment, and the use of older adult mice allows time for one or two rounds of vaccination prior to challenge. The model provides read-outs of both infection (viraemia and viral RNA levels in the livers) and disease (weight loss, liver lesions and liver histopathology). Limitations include the inconsistent penetrance of overt liver lesions, as well as no elevation in bilirubin levels and high variability in elevated serum AST levels (although the latter does recapitulate the variability seen in human YFV infections). As illustrated herein, the model can be used to assess vaccine efficacy, but it may also find utility for other studies involving, for instance, fetal infection/development [48,49], immunopathology [50] and/or tropism [51].

This paper describes the preclinical efficacy of the third flavivirus vaccine to be based on the chimeric Binjari technology, after the Zika virus vaccine, BinJ/ZIKV-prME [24], and the West Nile virus vaccine, BinJ/WNV-prME [25], with another paper for a dengue vaccine (BinJ/DENV-prME) in preparation. Taken together these data illustrate the broad utility of the Binjari platform for generating flavivirus vaccines.

Herein neutralizing antibodies were only generated in IFNAR-/- mice when BinJ/YFV-prME was formulated with adjuvant, whereas we previously reported that the BinJ/ZIKV-prME chimera induced neutralizing antibodies in these mice without the use of adjuvant [24]. Significant neutralizing antibody responses were, however, seen in wild-type outbred CD1 mice after 2 vaccinations with 5 µg of BinJ/YFV-prME in the absence of adjuvant (Appendix A). This perhaps supports the view that the viral RNA in these Binjari chimeras may provide some adjuvant activity [24,25], which might be expected to be less effective in IFNAR-/- mice. While unadjuvanted BinJ/WNV-prME vaccine also induced neutralizing antibodies in CD1 mice, the addition of the adjuvant Advax^TM^ also improved immune responses for this chimera [25]. YFV is a relatively distantly related flavivirus when compared with WNV and ZIKV [52] and YFV 17D may thus behave differently in, *inter alia*, neutralization assays, the mouse models, and with respect to protective correlates and/or pathological/immunopathological mechanisms. We have also not undertaken formal characterization of impurities and their levels in the different vaccine preparations, which may also influence their behaviors. Future work is seeking to establish GLP manufacture in C6/36 cells, characterize any inherent adjuvant activity, as well as exploring the potential for optimizing Binjari chimera vaccines by, for instance, modifying viral envelope protein sequences to present conformations more likely to induce neutralizing antibodies [53,54,55]. A clear goal would be to achieve protective immunity with a single vaccination, as is currently the case for the YFV 17D vaccine. 

This report describes preclinical efficacy data for a BinJ/YFV-prME vaccine in a mouse model of YFV 17D infection and disease and provides additional support for the utility of the Binjari virus technology for flavivirus vaccine development.

## 5. Conclusions

Adult female IFNAR-/- mice and the YFV-17D vaccine virus strain provide a model of YF infection and disease that recapitulates many aspects of human infection and disease. The model can be used for *inter alia* preclinical evaluation of YF vaccines, providing readouts of weight loss, viraemia and liver pathology. The Binjari virus technology for generating chimeric, VLP-like, flavivirus vaccines has utility for the generation of YF vaccines, although currently, adjuvant and two vaccinations are required to generate neutralizing antibodies and provide protection against YFV-17D challenge in IFNAR-/- mice.

## Figures and Tables

**Figure 1 vaccines-08-00368-f001:**
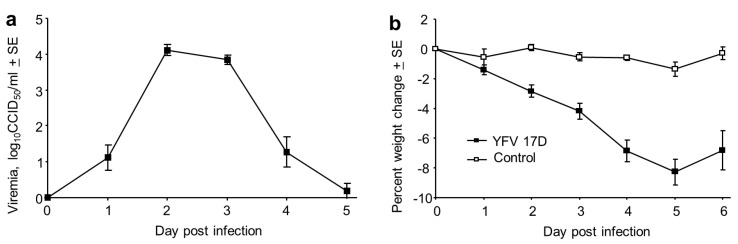
Viraemia and weight loss in IFNAR-/- mice infected with YFV 17D. (**a**) Viraemia after s.c. infection of male and female IFNAR-/- mice with 5 log_10_CCID_50_ of YFV 17D. Limit of detection 2 log_10_CCID_50_/mL for each mouse (*n* = 13 mice); viraemias below the level of detection were deemed to have a viraemia of 0 log_10_CCID_50_. Mean age 13.2 SD 3.2, range 12–20 weeks. (**b**) Percentage weight change for the mice in a, compared with 5 mice that were mock infected and also bled as for the infected mice (Control). Differences between infected and Control mice were significant on day 2 through to day 6 (*p* < 0.003, Kolmogorov Smirnov tests).

**Figure 2 vaccines-08-00368-f002:**
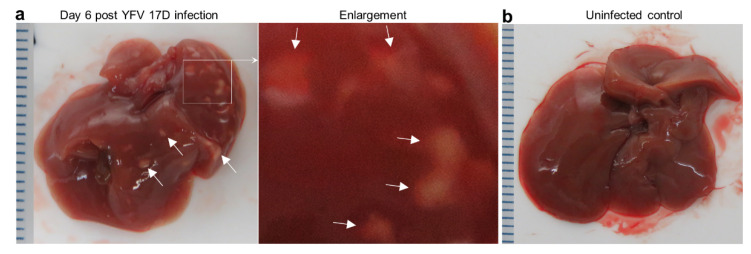
Liver lesions in YFV 17D-infected IFNAR-/- female mice. (**a**) Overt white necrotic lesions visible in the liver of an infected mouse day 6 post infection (white arrows). (**b**) A liver of an uninfected mouse with no white lesions. (**c**) H & E of necrotic liver lesion in YFV 17D-infected mouse. Paucity of blue nuclei evident in the necrotic lesion (pink), with inflammatory infiltrates (blue nuclei) at the periphery. (**d**) IHC of parallel section (from c) stained with the anti-flavivirus monoclonal antibody 4G4. Staining (brown) is clearly visible at the periphery of the necrotic lesion and in inflammatory foci (top left and right). Light counterstaining with haematoxylin (blue) show cell nuclei. (**e**) IHC staining with a control monoclonal antibody. (**f**) Councilman bodies (black dotted ovals) observed by H & E staining of livers from four YFV 17D-infected mice. (**g**) Top; steatosis (accumulation of lipid droplets in hepatocytes) visible as small white rounded inclusions in a liver of a YFV 17D-infected mouse. Bottom; uninfected control liver.

**Figure 3 vaccines-08-00368-f003:**
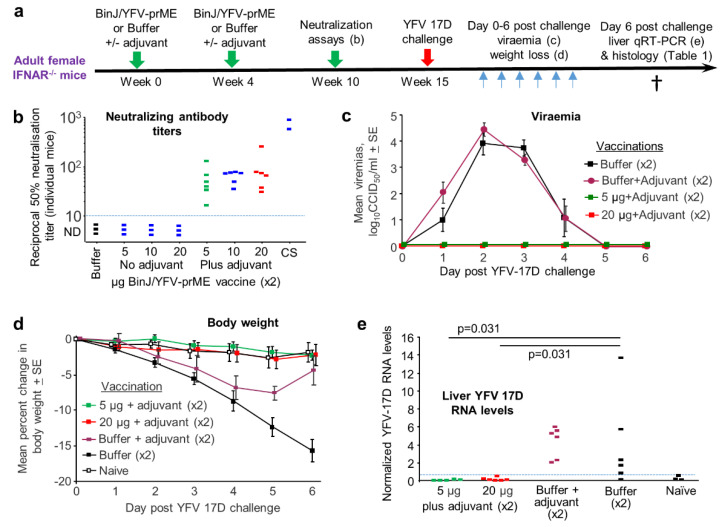
Evaluation of the BinJ/YFV-prME vaccine in female IFNAR-/- mice. (**a**) Time course of experiment. (**b**) Neutralizing antibody responses for No adjuvant and Plus adjuvant groups with the indicated dose of vaccine (µg) given twice (×2). CS–pooled convalescent serum taken from IFNAR-/- mice infected 6–10 weeks previously with YFV 17D. Limit of detection was a dilution of 1 in 10 (dotted line). (**c**) Three groups from b (indicated by black, green and red squares) were challenged with YFV 17D and viraemia post challenge shown (limit of detection for each mouse is 2 log_10_CCID_50_/mL). (**d**) Percent weight loss for the mice in c. A fourth group is included (*n* = 5) who were bled daily as for the infected mice, but that received no vaccination or challenge. Differences between Buffer and BinJ/YFV-prME vaccinated mice were significant on days 3 through 6 (*p* < 0.01, t tests). (**e**) qRT PCR for YFV 17D in livers taken day 6 post challenge in the indicated groups. Horizontal line indicates cut-off for reliable detection, with multiple repeat tests of Naïve livers falling below this line. Statistics by Kolmogorov Smirnov tests.

**Table 1 vaccines-08-00368-t001:** Histological features of mouse liver H & E stained sections day 6 post YFV 17D challenge (see Figure 3. Enumeration is per liver section. Distances (in µm) represent the maximum distance (edge to edge) for each lesion, or range of distances for the indicated number of lesions.

**Buffer**	
Mouse 1	Councilman bodies. 15 infiltrate foci (16–52 µm). Marked steatosis
Mouse 2	Councilman bodies. 29 infiltrate foci (10–114 µm). Mild steatosis
Mouse 3	Councilman bodies. 22 infiltrate foci (20–52 µm). Mild steatosis
Mouse 4	3 infiltrate foci (15–34 µm)
Mouse 5	*Necrotic lesion* (325 µm). Councilman bodies. 17 infiltrate foci (17–104 µm). Mild steatosis
Mouse 6	*Necrotic lesions* (212 and 43 µm). Councilman bodies. 18 infiltrate foci (15–52 µm). Mild steatosis
**Buffer + adjuvant**	
Mouse 1	Councilman bodies. 6 infiltrate foci (18–27 µm). Small patches of steatosis.
Mouse 2	*Necrotic lesion* (105 µm). Councilman bodies. 16 infiltrate foci (22–54 µm)
Mouse 3	Councilman bodies. 16 infiltrate foci (21–63 µm)
Mouse 4	*Necrotic lesions* (342 and 197 µm). Councilman bodies. 14 infiltrate foci (16–108 µm). Areas of marked steatosis
Mouse 5	Councilman bodies. 22 infiltrate foci (17–109 µm). Small patches of steatosis
Mouse 6	*Necrotic lesions* (357 and 123 µm). Councilman bodies. 30 infiltrate foci (21–79 µm)
**5 µg** **BinJ/YFV-prME + adjuvant**	
Mouse 1	7 infiltrate foci (29–53 µm)
Mouse 2	6 infiltrate foci (15–38 µm)
Mouse 3	11 infiltrate foci (12–42 µm)
Mouse 4	11 infiltrate foci (13–77 µm)
Mouse 5	6 infiltrate foci (18–82 µm)
Mouse 6	7 infiltrate foci (18–34 µm)
**20 µg** **BinJ/YFV-prME + adjuvant**	
Mouse 1	5 infiltrate foci (14–28 µm)
Mouse 2	5 infiltrate foci (21–50 µm)
Mouse 3	5 infiltrate foci (24–44 µm)
Mouse 4	5 infiltrate foci (19–35 µm)
Mouse 5	6 infiltrate foci (19–139 µm)
Mouse 6	3 infiltrate foci (26–43 µm)
**Naïve mice (no vaccine, no challenge)**	
Mouse 1	2 infiltrate foci (24–31 µm)
Mouse 2	3 infiltrate foci (31–69 µm)
Mouse 3	2 infiltrate foci (26–69 µm)
Mouse 4	1 infiltrate foci (32 µm)
Mouse 5	3 infiltrate foci (11–22 µm)
Mouse 6	0 infiltrate foci

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
