# Peer review of "A Yellow Fever Virus 17D Infection and Disease Mouse Model Used to Evaluate a Chimeric Binjari-Yellow Fever Virus Vaccine"

_vaccines, 2020, doi:10.3390/vaccines8030368_

Round 1

Reviewer 1 Report

The paper describes some interesting experimental work that is of importance. Some improvements could be made.

The introduction is somewhat short and the description of the proteins is somehat slipshod. Proteins are crucial aspects in all vaccines. If something is wrong with a vaccine, it always comes down to the proteins.

* A brief description needs to be written for each of the proteins involved. References for these can be found in any medical microbiology or virology text.

* I noticed that the experiment does not include tests on pregnant mice. A study by Goh et al have shown correlations between outer shell (M and PrM) disorder in dengue and Zika strains to fetal morbidy:

Goh GK, Dunker AK, Foster JA,Uversky VN. Zika and flavivirus disorder: Virulence and

fetal morbidity. Biomolecules. 2019b;9: e710.

YFV PrM and M are very disordered and there are very few studies on YFV and its vaccine as related to fetal morbidty.. While I understand that this may be outside the scope of the paper, I do think this could be a good experimental study for the next research. I also think that the authors should place a warning somewhere in the paper that this issue has been poorly studied.

* This manuscript on chimera vaccine should contain a passage on the possible mechanism of action fo the vaccine

I noticed that in the vaccine chimera, the PrM is from the whild-type but the C is from a totally different virus, According to Goh et al, C disorder is correlated with mortality in ALL flaviviruse and among becuse the higher disorder ensures greater efficient binding via protein-protein interactions and thus causing rapid replcation ie higher viral load before the host immune system recognizes it. And YFV is highly virulent because its C, the capsid, is very disordered, In the case of the chimera vaccine, the C has been completely changed presumably to a much more ordered C. Goh et al may have described possible mechanism of the vaccine.

.

It is also possible that chimera vaccine causes such low viral load that the virus does not penetrate the placenta where the fetus is but that needs to be seen in later experiments on pregnant mice.

Other minr correstions that need to be made are:

“supernatant removed and the virus 105precipitate suspended in 10 mM Tris pH 8, 1 mM EDTA and 120 mM NaCl”

should be changed to

“...with the supernant removed...”

“cell debris removed by 30 min”

should be changed to

“with cell debris removed...”

The supernatant and sucrose was discarded

should be be corrected to

“..were discarded”

“There were 2.6 fold”

should be corrected to

“..2.6 folds..”

Author Response

REVIEWER 1

1) The introduction is somewhat short and the description of the proteins is somewhat slipshod. Proteins are crucial aspects in all vaccines. If something is wrong with a vaccine, it always comes down to the proteins. * A brief description needs to be written for each of the proteins involved. References for these can be found in any medical microbiology or virology text.

RESPONSE: The introduction has been substantially expanded to describe more precisely the vaccine prME proteins, and the composition of the vaccine.

2) I noticed that the experiment does not include tests on pregnant mice. A study by Goh et al have shown correlations between outer shell (M and PrM) disorder in dengue and Zika strains to fetal morbidy: Goh GK, Dunker AK, Foster JA,Uversky VN. Zika and flavivirus disorder: Virulence and fetal morbidity. Biomolecules. 2019b;9: e710.

YFV PrM and M are very disordered and there are very few studies on YFV and its vaccine as related to fetal morbidty.. While I understand that this may be outside the scope of the paper, I do think this could be a good experimental study for the next research. I also think that the authors should place a warning somewhere in the paper that this issue has been poorly studied.

RESPONSE: The issue of fetal morbidity is clearly of importance, but has recently been addressed in a mouse model (da Silva et al Yellow Fever Vaccination in a Mouse Model Is Associated With Uninterrupted Pregnancies and Viable Neonates Except When Administered at Implantation Period.  Front. Microbiol., 2020 Epub) [1].  We have added this reference and the Goh reference [2] to the manuscript in the Discussion to highlight other potential applications of our YFV model. (Not sure our study justifies a warning as such).

3)  This manuscript on chimera vaccine should contain a passage on the possible mechanism of action of the vaccine

RESPONSE: We have added the sentence “Neutralizing antibodies directed at E are believed to be the primary mediators of protection for most flavivirus vaccines, including YFV vaccines [3] [4] to the introduction.

4) I noticed that in the vaccine chimera, the PrM is from the wild-type but the C is from a totally different virus, According to Goh et al, C disorder is correlated with mortality in ALL flaviviruses and among because the higher disorder ensures greater efficient binding via protein-protein interactions and thus causing rapid replication i.e. higher viral load before the host immune system recognizes it. And YFV is highly virulent because its C, the capsid, is very disordered, In the case of the chimera vaccine, the C has been completely changed presumably to a much more ordered C. Goh et al may have described possible mechanism of the vaccine.

RESPONSE: We are in the process of undertaking a series of capsid swap experiments, which are yet to be completed and it is probably inappropriate to speculate until the data has been collected.  However, the issue of virulence in mice cannot be addressed using the Binjari chimeras as they do not replicate in vertebrates. 

5) It is also possible that chimera vaccine causes such low viral load that the virus does not penetrate the placenta where the fetus is but that needs to be seen in later experiments on pregnant mice.

RESPONSE: Our experiments have illustrated that the chimeric vaccines do not replicate at all in mice (see Hobsen-Peters et al 2019 Fig. S2).  However, placental infection by the chimeric vaccine may be worth testing as a safety issue when this technology is further down the development pathway.

6) All minor correction have been made except the 2.6 fold(s).  The latter is correct in the original.

References

  1. da Silva FC, Magaldi FM, Sato HK, Bevilacqua E. Yellow Fever Vaccination in a Mouse Model Is Associated With Uninterrupted Pregnancies and Viable Neonates Except When Administered at Implantation Period. Front Microbiol. 2020;11: 245.
  2. Goh GK, Dunker AK, Foster JA, Uversky VN. Zika and Flavivirus Shell Disorder: Virulence and Fetal Morbidity. Biomolecules. 2019;9.
  3. Hurtado-Monzon AM, Cordero-Rivera CD, Farfan-Morales CN, Osuna-Ramos JF, De Jesus-Gonzalez LA, Reyes-Ruiz JM, et al. The role of anti-flavivirus humoral immune response in protection and pathogenesis. Rev Med Virol. 2020: e2100.
  4. Davis EH, Barrett ADT. Structure-Function of the Yellow Fever Virus Envelope Protein: Analysis of Antibody Epitopes. Viral Immunol. 2020;33: 12-21.

Reviewer 2 Report

In the article “A Yellow Fever Virus 17D Infection and Disease Mouse Model Used to Evaluate a Chimeric Binjari-Yellow Fever Virus Vaccine,” Yan and colleagues continue the expansion of the use of the insect-specific flavivirus, Binjari virus, as a platform for the development of flavivirus vaccines.  In the present article, they test a yellow fever virus (YFV) construct for protection from diseases in IFNAR-/- mice challenged with YFV 17D.  As described in the paper, the model produces little overt clinical disease, but does yield measurable disease indicators (especially weight loss), as well as characteristic histologic lesions in the liver evident upon sacrifice.  The age and gender bias observed is interesting, but is beyond the scope of this manuscript.  The model has the advantage of using the 17D strain of YFV in the challenge, so experiments can be performed at BSL-2.

The authors bring to this paper considerable experience and expertise in all areas represented in this manuscript:  arbovirology, flaviviruses, vaccines, etc.  The manuscript is quite well written and well presented.  Experimentally, the paper is complete.  The conclusions are supported by the data. 

While the authors mention the advantages of the Binjari virus platform over the current state of the art, one disadvantage is worth mentioning.  The chimeric virus used here requires two doses while the live attenuated vaccines currently used only require a single dose.  For a vaccine to be most viable in an outbreak response setting, single-dose efficacy is important.  As this does have implications for the further development of this YFV vaccine, a mention is the discussion is suggested.

   A couple of specific minor points are presented below:

   Line 96-98:  This is a fairly long run-on sentence.  For ease of reading suggest breaking it up.

   Line 168 (figure 1a):  If the LOD is 2 log10 CCID50, as is indicated in the legend (line 170-171), it is not clear how the 1, 4, and 5 dpi datapoints can be approximately 1, 1.2, and 0.2, respectively.  How is that calculated when it is not known whether the undetectable samples are zero or 1.9?

Author Response

REVIEWER 2

1) While the authors mention the advantages of the Binjari virus platform over the current state of the art, one disadvantage is worth mentioning.  The chimeric virus used here requires two doses while the live attenuated vaccines currently used only require a single dose.  For a vaccine to be most viable in an outbreak response setting, single-dose efficacy is important.  As this does have implications for the further development of this YFV vaccine, a mention is the discussion is suggested.

RESPONSE: We have added a sentence to the Discussion to address this issue.  “A clear goal would be to achieve protective immunity with a single vaccination as is currently the case with the YFV 17D vaccine”.

2) Line 96-98:  This is a fairly long run-on sentence.  For ease of reading suggest breaking it up.

RESPONSE: This has been rephrased to improve clarity.

3) Line 168 (figure 1a):  If the LOD is 2 log10 CCID50, as is indicated in the legend (line 170-171), it is not clear how the 1, 4, and 5 dpi datapoints can be approximately 1, 1.2, and 0.2, respectively.  How is that calculated when it is not known whether the undetectable samples are zero or 1.9?

RESPONSE: We have added the sentence to the figure legend “viraemias below the level of detection were deemed to have a viraemia of 0 log10CCID50”to clarify this issue.  This is standard practice in the field and results in a conservative estimate of the actual viremias; whereas, using 1.9 would likely often lead to an over-estimate of the viremias.